# Phylogenetic Diversity of Ossification Patterns in the Avian Vertebral Column: A Review and New Data from the Domestic Pigeon and Two Species of Grebes

**DOI:** 10.3390/biology11020180

**Published:** 2022-01-24

**Authors:** Tomasz Skawiński, Piotr Kuziak, Janusz Kloskowski, Bartosz Borczyk

**Affiliations:** 1Department of Palaeozoology, Faculty of Biological Sciences, University of Wrocław, Sienkiewicza 21, 50-335 Wrocław, Poland; 2Department of Evolutionary Biology and Conservation of Vertebrates, Faculty of Biological Sciences, University of Wrocław, Sienkiewicza 21, 50-335 Wrocław, Poland; 326967@uwr.edu.pl (P.K.); bartosz.borczyk@uwr.edu.pl (B.B.); 3Department of Zoology, Institute of Zoology, Poznań University of Life Sciences, Wojska Polskiego 71C, 60-625 Poznań, Poland; januszkl@up.poznan.pl

**Keywords:** axial skeleton, birds, evo-devo, ossification sequences, skeletogenesis

## Abstract

**Simple Summary:**

There are still many unknowns in the development of the skeleton in birds. Traditionally, the neck vertebrae were considered to be the first ossifying elements in the spine. Later studies have shown that this is not always the case. In some species, the thoracic vertebrae ossify even before them. Evolutionary analyses indicate that ancestrally the spine starts ossifying from two different sites, one located in the neck, the other in the thorax. However, the Neoaves, a group that includes all living birds except the palaeognaths, landfowl and waterfowl, are very poorly studied. In this article, we review the information about ossification patterns of the spine in birds. We also describe its development in three neoavians, the pigeon and two grebes. In the pigeon, the neck vertebrae were the first to ossify, but in the grebe, the thoracic vertebrae ossified earlier. Our analyses confirm the ancestral presence of two sites from which the ossification of the spine starts in birds.

**Abstract:**

Despite many decades of studies, our knowledge of skeletal development in birds is limited in many aspects. One of them is the development of the vertebral column. For many years it was widely believed that the column ossifies anteroposteriorly. However, later studies indicated that such a pattern is not universal in birds and in many groups the ossification starts in the thoracic rather than cervical region. Recent analyses suggest that two loci, located in the cervical and thoracic vertebrae, were ancestrally present in birds. However, the data on skeletal development are very scarce in the Neoaves, a clade that includes approximately 95% of extant species. We review the available information about the vertebral column development in birds and describe the ossification pattern in three neoavians, the domestic pigeon (*Columba livia domestica*), the great crested grebe (*Podiceps cristatus*) and the red-necked grebe (*Podiceps grisegena*). In *P. cristatus*, the vertebral column starts ossifying in the thoracic region. The second locus is present in the cervical vertebrae. In the pigeon, the cervical vertebrae ossify before the thoracics, but both the thoracic and cervical loci are present. Our ancestral state reconstructions confirm that both these loci were ancestrally present in birds, but the thoracic locus was later lost in psittacopasserans and at least some galloanserans.

## 1. Introduction

The vertebral column is a defining feature of vertebrates. It consists of a series of segmental units, the vertebrae, which play a crucial role in vertebrate biology, such as protecting the spinal cord or providing muscle attachment sites for numerous muscles involved in locomotion. The vertebra itself is usually a complex structure composed of several distinct parts, the body, arches and a few projections, such as spinous or transverse processes (e.g., [1]). The vertebral column is divided into several regions, the cervical, thoracic, lumbar, sacral and caudal, which differ morphologically (e.g., [2]). There is also a great interspecific variation in the number of vertebrae that constitute the column. However, the number of vertebrae is constrained at relatively low numbers in mammals, while in sauropsids (reptiles and birds) these numbers are usually higher and much less conservative [3,4]. In addition, the vertebral column is strongly modified in extant birds as an adaptation to active flight; many of the individual vertebrae fuse (usually after hatching [5,6,7,8]) to form compound structures—the notarium, synsacrum and pygostyle (e.g., [2,6,7,9]). Although the morphology and number of vertebrae can tell us a lot about developmental processes which act on the vertebral column (e.g., [3,10,11]), the data on the sequence in which it ossifies are still incomplete for many groups.

The ossification sequences—the order in which individual bones appear in development—are one of the most basic, yet important studies in developmental biology. They give us not only information about the development of a given species but also allow us to compare different species, thus placing these sequences in an evolutionary context. They also help us interpret fossil embryos and reconstruct ancestral developmental patterns and heterochronic events (e.g., [12,13,14,15]). With over 10,000 extant species, the birds are one of the most species-rich lineages of tetrapods. They exhibit great ecological diversity and occupy numerous different niches and habitats. Their life histories are also very diverse—some birds, such as galliforms (Galliformes), including the well-studied chicken and quail, are precocial, i.e., have independent hatchlings, while many others are altricial, with embryo-like hatchlings. Numerous species occupy intermediate positions in this precocial-altricial spectrum (e.g., [16]). Despite this huge diversity, the temporal order of bone ossification in birds has been described as remarkably conservative, with only small differences between species widely divergent ecologically and phylogenetically [16]. In particular, the development of the vertebral column has been repeatedly cited as predominantly or even universally proceeding from anterior to posterior in birds [12,16,17,18]. Although further studies indicated a greater variation in this respect among birds (e.g., [19,20,21,22]), ossification patterns other than the anteroposterior are regarded as exceptions [23]. Recently, there has been an increase in knowledge on the genetic and developmental bases of the regionalisation of the vertebral column in birds (e.g., [10,11,24]). However, despite the long history of studies on the development of the avian skeleton (e.g., [25,26]), the data on the ossification sequences are still scarce and limited mostly to precocial birds, in particular palaeognaths and galloanserans, while the information about neoavians concerns only a few species (e.g., [13]). This is probably related to the fact that a much greater proportion of neoavians represents altricials and in many altricial species (although this is far from being universal) the vertebral column starts ossifying only after hatching (e.g., [16,18,19]). In general, the postnatal skeletal ontogeny of birds is very poorly known (e.g., [8]), so it hampers our understanding of the development and evolution of the vertebral column in many groups of birds. In this article, we attempt to summarise currently available information about the ossification patterns in extant birds and supplement it with new data from two basal neoavian lineages, the altricial (‘altricial 1′ according to Starck [16]) pigeons (Columbiformes) and precocial (‘precocial 3′ in the classification by Starck [16], which means that they are relatively little precocial) grebes (Podicipediformes) (relationships of the species discussed in the text are shown in Figure 1). We focused on the following three different processes: (1) body ossification, (2) vertebral arch ossification and (3) vertebral arch fusion, with a particular reference to the first two.

## 2. Materials and Methods

### 2.1. Literature Review

The information given below are summarised in Table 1 and Figure 2, Figure 3 and Figure 4. The exact individual age of the studied birds was often unknown in the publications discussed below (as well as in the specimens described in the ‘Results’), so the timing of the observed events is often not directly comparable, but we were interested primarily in the sequence in which these events appear during the ontogeny, which can be compared even without information about the individual age of the specimens.

#### 2.1.1. Vertebral Bodies (Corpora Vertebrae)

In palaeognathous birds, the ossification of the vertebral bodies (and the vertebral column in general) usually starts with the thoracic vertebrae. This is the condition observed in most of the hitherto studied species, including both the flying tinamous (Tinamidae) and secondarily flightless ratites. In the ostrich (*Struthio camelus*), the bodies of the thoracic and preacetabular synsacral (which belong to the lumbar region) vertebrae are the first to ossify, usually in 23-day-old embryos, but in some individuals, slightly later. In 25-day-old embryos, the cervical vertebrae ossify, but the ossification of the first two cervicals, the atlas and axis, is delayed and starts in 28-day-old embryos. The free caudal vertebrae start to ossify in 30-day-old embryos (although there is some variability in this respect) and finally the bodies of the vertebrae that form the pygostyle ossify in 37-day-old embryos [30]. A very similar condition is observed in the elegant crested tinamou (*Eudromia elegans*). The bodies of the posterior thoracics and anterior synsacrals (lumbars) are the first to appear in 11-day-old embryos. From there, the ossification spreads anteriorly and posteriorly and in 14-day-old embryos, all cervical and synsacral bodies had begun ossifying [30]. In the greater rhea (*Rhea americana*), the bodies of the thoracic vertebrae 3–6 are the first to ossify in stage 37 (according to Hamburger and Hamilton [38]) embryos. In late stage 38, the cervical vertebrae start ossifying. In 22-day-old embryos (stage 40+), all thoracic vertebrae are ossified. The emu (*Dromaius novaehollandiae*) differs from other studied palaeognaths in the earlier onset of ossification in the cervical vertebrae bodies. In stage 38 embryos, the ossification starts from two loci (*sensu* Verrière et al. [13]), one located in the cervical vertebrae 2–4 (C2–C4) while the second one contains thoracic vertebrae located posteriorly to the thoracic 2 (T2). A few synsacral bodies located under anterior parts of the ilia are also ossified. In the following stage (39), all cervicals except the atlas, as well as all thoracics and synsacrals are ossified. In stage 40+ embryos, the free caudals and, lastly, the pygostyle, start ossifying [30]. In his study on the anatomy and development of the kiwi, Parker [25] described a specimen (“stage G”) of the little spotted kiwi (*Apteryx owenii*), in which the bodies of almost all cervical (except the atlas), thoracic, lumbar and sacral vertebrae had already begun their ossification, while the caudals were still cartilaginous. It is, however, unclear whether all these vertebrae indeed ossify nearly simultaneously or whether it is an artefact of an insufficient number of studied embryos.

The galloanserans differ from palaeognaths in that they usually conform to the ‘typical’ anteroposterior gradient of ossification. In the turkey (*Meleagris gallopavo*), the ossification seems to proceed in a strictly anterior to posterior order, with the most anterior cervical vertebrae ossifying at stage 38. The thoracic bodies start ossifying at stage 40+ and are followed by the ossification centres in the synsacral (lumbar) vertebrae. Lastly, ossifications appear in the caudal vertebrae [21,31]. The chicken (*Gallus domesticus*) is similar to the turkey in the order of ossification of the vertebral centra; in general, the cervical bodies are followed by the thoracic, then synsacral and finally caudal vertebrae [18]. However, in contrast to the turkey, but similarly to most palaeognaths, the ossification of the anteriormost cervicals is delayed [17,21]. The ossification of the most posterior cervical precedes the more anteriorly located cervical vertebrae; from there, the ossification proceeds posteriorly [21], so the last cervical belongs to the thoracic locus (*sensu* Verrière et al. [13]). The ossification of the vertebral column in the Japanese quail (*Coturnix japonica*) seems to be more similar to the chicken than to the turkey. The mid-cervical vertebrae (6–9) ossify first and the ossification spreads both anteriorly and posteriorly [21,32,33]. A similar condition is present in the thoracic vertebrae, in which mid-thoracics appear first and the ossification proceeds both anteriorly and posteriorly. Anterior synsacrals ossify next, followed by posterior synsacrals, then by free caudals and lastly by the pygostyle [32,33]. Interestingly, Blom and Lilja [19] described a strictly anteroposterior order of ossification in *C. japonica* which may suggest an intraspecific variability in the ossification patterns. A closely related blue-breasted quail (*Synoicus chinensis*) differs from the Japanese quail in that the ossification starts with anterior cervicals and all thoracic vertebrae (11-day-old embryos). They are followed by the remaining cervicals and anterior synsacrals (12-day-old), mid-synsacrals (13-day-old), posterior synsacrals and anterior caudals (14-day-old) and finally by the remaining caudals (17-day-old) [34]. In the Muscovy duck (*Cairina moschata*), all precaudal vertebrae ossify simultaneously (stage 39), without a gradient, and are followed closely by the first three caudal bodies [20]. In the common eider (*Somateria mollissima*), the cervical bodies follow an anteroposterior gradient. Posterior cervicals are worse ossified than the anterior thoracics which suggests that the ossification proceeds from two loci. Ten anterior synsacrals also ossify at the same stage (38). In stage 39 the remaining synsacrals ossify and are followed (stage 40) by the caudal vertebrae [20]. The vertebral bodies ossify in an anteroposterior direction in the mallard (*Anas platyrhynchos*). The cervicals appear first (stage 37) and are followed by thoracics and approximately half of synsacrals (stage 39). In stage 40 free caudal bodies ossify, then the remaining synsacrals and, finally, the anterior components of the pygostyle [20].

The neoavians, which represent approximately 95% of all extant species of birds [27], are relatively the most poorly studied avian clade. In the common tern (*Sterna hirundo*) the ossification of the vertebral column starts with the thoracic bodies. The ossification proceeds posteriorly from the first thoracic. Later, the ossifications in the cervicals appear. Anterior synsacral bodies ossify approximately at the same time, with the more posteriorly located ones ossifying later. The caudal bodies appear last [22]. In the great skua (*Stercorarius skua*) the cervical and thoracic bodies ossify approximately at the same time and are followed closely by the synsacral bodies. The bodies of the caudals appear last [35]. In the black-headed gull (*Chroicocephalus ridibundus*), the cervical and thoracic bodies ossify nearly simultaneously [22,37], but Rogulska [36] described that they precede the thoracics, which may indicate an intraspecific variation or an insufficient sampling in the work of Schumacher and Wolff [37]. In the common gull (*Larus canus*) the thoracic bodies were described to ossify slightly before the cervicals [22,37]. In both these species, the remaining vertebrae were not described, so they must ossify later. The monk parakeet (*Myiopsitta monachus*) exhibits a typical anteroposterior gradient of ossification. The cervical vertebrae ossify first (although the appearance of the atlas is delayed), the thoracic vertebrae secondly, and the synsacral and caudal vertebrae lastly [14]. There is some intraspecific variation in the timing of the ossification of the vertebral column in the specimens of the zebra finch (*Taeniopygia guttata*) studied by Mitgutsch et al. [12], but in most of them, the direction of ossification proceeded from anterior to posterior, with the cervical vertebrae ossifying first, thoracics secondly and synsacrals and caudals thirdly [12]. An anteroposterior order of ossification was also described for the rook (*Corvus frugilegus*), in which the cervical vertebrae appear first during the development [36].

#### 2.1.2. Vertebral Arches (Arcus Vertebrae)

The ossification of the vertebral arches usually takes place after the appearance of the bodies of the respective vertebrae. The exception is the atlas, in which the arches ossify before the body. This condition is present in many species of phylogenetically distant species (e.g., [17,30,33]), so probably represents an ancestral condition. Curiously, in the Eurasian reed warbler (*Acrocephalus scirpaceus*), the atlantal body was described to be absent, and the atlas is composed only of two arches which fuse dorsally and ventrally [8].

The ossification of the vertebral arches is much less variable in birds than the ossification patterns of the bodies. Usually, it much more closely conforms to the anteroposterior order, even if the bodies ossify with a different pattern. In *Struthio camelus*, the cervical arches ossify first (in 28-day-old embryos), with the atlantal and axial arches being delayed, and are completely ossified in 31-day-old embryos. The thoracic arches ossify at the same time or slightly later. The synsacral arches ossify next, usually in 36-day-old embryos, but their ossification may be slightly delayed in some individuals [30]. *Rhea americana* and *Dromaius novaehollandiae* follow a similar pattern, with the cervical arches ossifying first (in stage 40+, 22-day-old embryos in the former species and at stage 39 in the latter). In *D. novaehollandiae*, the atlas and axis ossify first, the remaining cervicals and thoracics appear next (stage 40+, 36-day-old embryos). The ossification of the synsacral arches is variable but takes place much later (approximately in 41-day-old embryos). In *R. americana* a clear order of ossification in cervical arches was not described and the atlas does not seem to be better ossified than the remaining cervicals. Thoracic arches begin ossifying later (stage 40+, 28-day-old embryos) and are followed by the synsacral arches (approximately 30-day-old embryos) [30]. In *Eudromia elegans*, the arches begin ossifying simultaneously with the bodies in the cervicals. There are two loci within the cervical series, one located in the atlas and the other in C4 and more posteriorly located vertebrae (14-day-old embryos). The thoracic arches ossify slightly later (15-day-old embryos) [30].

In *Gallus domesticus*, the vertebral arches ossify first in the cervicals (C2–C9) in 14-day-old embryos. The ossification in the atlas is delayed and takes place approximately two days later. In 19-day-old embryos, the arches are ossified also in thoracic and synsacral vertebrae [17]. The caudal arches ossify last [21]. In *Meleagris gallopavo*, the cervical arches are also the first to appear (at stage 40+, 18-day-old embryos), simultaneously with the arches of the thoracic vertebrae located anteriorly to the notarium. In 20-day-old embryos, all thoracic arches are ossified. The caudal arches ossify in 23-day-old embryos [21]. In *Synoicus chinensis*, the vertebral arches of the cervicals and anterior thoracics are ossified in 12-day-old embryos. They are followed by the ossifications in mid-thoracics (13-day-old), posterior thoracics, anterior synsacrals (lumbosacrals) and anterior caudals (14-day-old), mid- and posterior synsacrals and mid-caudals (15-day-old) and, finally, posterior caudals (17-day-old) [34]. There are several loci in vertebral arches in *C. japonica*. The atlantal arches are the first to ossify (in 10-day-old embryos) and are followed by ossifications in the axis and cervicals 6–9 (11-day-old embryos), then by cervicals 3–5, thoracic 1 and synsacrals 1–3 (12-day-old embryos), then by cervicals 10–13, thoracics 2–6 and synsacrals 4–7 (13-day-old embryos), then by cervicals 14–15, synsacrals 8–12 and caudals 1–6 (14-day-old embryos) and, lastly, by four vertebrae that form the pygostyle (15-day-old embryos) [33]. In *Anas platyrhynchos*, the vertebral arches ossify in a typical, anteroposterior sequence (at stage 40+). The cervical arches ossify first, anterior thoracics (T1–T2) come next and are followed by the remaining thoracics, then by synsacral arches and, finally, by caudal arches [20]. In *Somateria mollissima*, the sequence is very similar, the only difference is that anterior thoracics (T1–T2) ossify simultaneously with cervicals (stage 39). *Cairina moschata* also develops very similarly. The cervical arches appear first (stage 40+), preceding the ossification of the anteriormost thoracic arch (T1). In contrast to previous two species, the caudal arches ossify slightly before the synsacral arches [20].

In *Sterna hirundo*, the ossification of the vertebral arches starts variably either with the cervicals or with the thoracics. In the cervicals, the locus appears to be present in mid-cervicals and the ossification proceeds both anteriorly and posteriorly. In thoracic vertebrae, the more posteriorly located arches are more poorly ossified which suggests and anteroposterior order of ossification in the thoracics. Later, the synsacrals and the anteriormost two caudals ossify [22]. The knowledge of other charadriiforms is incomplete but the ossification of the vertebral arches seems to start with the cervicals in *Stercorarius skua* and *Chroicocephalus ridibundus* [22]. In *Myiopsitta monachus*, the cervical arches ossify first (stage 40+) and are followed by ossification of the synsacral and caudal vertebrae (after hatching). The ossification of the thoracic arches was not described but it probably takes place after cervical arches [14]. Although the ossification of the arches was not explicitly described for *Taeniopygia guttata* by Mitgutsch et al. [12], it was stated that “more anterior parts of the axial skeleton ossify before the more caudal groups” [12].

#### 2.1.3. Vertebral Arch Fusion

The knowledge of the timing and sequence of fusion of the vertebral arches is very incomplete in amniotes in general and in birds in particular [13]. In *Sterna hirundo*, the locus is present in mid-cervical vertebrae and the arches proceed to fuse bidirectionally [19]. The same pattern is present in *Chroicocephalus ridibundus* [8]. Verrière et al. [13] also described the presence of such locus in cervical vertebrae in *Phasianus*. The situation is more complex in the Eurasian reed warbler (*Acrocephalus scirpaceus*), with the presence of several such loci. The vertebral arches start to fuse soon after hatching. In an approximately 1-day-old hatchling the arches are fused in both the anterior cervical and posterior cervicals. The ossification starts with anterior part of the arch. The other locus is present in the thoracic vertebrae, with the arches of T4 and T5 being fused (the fusion is more advanced in the latter vertebra). In T6, the arches contact each other anteriorly but did not yet start to fuse [8].

### 2.2. Skeletal Development

We studied the development of the vertebral column in three species of neoavians, the great crested grebe (*Podiceps cristatus*), the red-necked grebe (*Podiceps grisegena*) and the homing pigeon, a variety of the domestic pigeon (*Columba livia domestica*). The sample of *P. cristatus* consisted of 12 perinatal specimens. Seven of these specimens represent late embryos with unknown date and place of collection. They were stored in 70% ethanol. The remaining five perinates were collected in between 1995 and 2004 in eastern Poland and later frozen. Because all of them were collected in the field, their exact individual age is not known. However, all embryos belong to 39+ stage according to the Hamburger and Hamilton [38] staging table for the chicken (*Gallus domesticus*). The specimens were described from the least ossified to the most ossified. For example, “ossification state 1” is the least ossified of the studied specimens. The ossification states are not exactly comparable between species, so the “ossification state 1” in *P. cristatus* represents a different stage than “ossification state 1” in *P. grisegena*. The sample of *P. grisegena* consisted of 18 perinatal specimens of unknown exact individual age, collected in the vicinity of Lublin (eastern Poland) in 1995–2013 and later frozen. The sample of *C. livia domestica* consisted of 5 neonatal specimens, collected in approximately 24 h intervals, ordered from the youngest to the oldest. All these specimens are currently in the collection of the Department of Evolutionary Biology and Conservation of Vertebrates, University of Wrocław (IZK).

Double-staining of the specimens followed the procedure described by Dingerkus and Uhler [39], with slight modifications. In brief, the specimens were stained for the presence of cartilage in alcian blue solution (with a reduced amount of glacial acetic acid and shorter period of staining than in the original procedure, to minimise the risk of decalcification), digested in pancreatin solution, stained for the presence of calcifications with alizarin, cleared in a growing series of potassium hydroxide-glycerin solutions and finally stored in 99% glycerin with the addition of thymol. The first uptake of alizarin (i.e., bone turning reddish) was regarded as the onset of ossification. The identification and nomenclature of anatomical structures follow primarily Baumel and Witmer [9] but using mostly the English equivalents rather than original Latin terms.

### 2.3. Ancestral State Reconstructions

We attempted to reconstruct the evolutionary history of the ossification patterns of the vertebral bodies in 21 species of birds (their relationships are shown in Figure 1 which includes also *Podiceps grisegena* and *Acrocephalus scirpaceus* which were not analysed). The higher-level relationships of birds are still contentious (see review in [27]), so the phylogenetic hypothesis used in this article is based on the ‘consensus phylogeny of birds’ from Braun and Kimball [27] while the interrelationships of the main clades follow Prum et al. [28] and Kuhl et al. [29]. Our goal was to reconstruct the number of loci in the vertebral column from which the bodies ossify (as did Verrière et al. [13] using a smaller sample of bird species). Because the ossification patterns of vertebral arch ossification were much less variable and the knowledge of vertebral arch fusion is very incomplete, we did not provide reconstructions for these characters. The analyses were conducted in Mesquite 3.61 [40], using the adopted phylogenetic framework (Figure 1) as a basis. We made the ancestral state reconstructions using maximum parsimony and maximum likelihood (Mesquite’s current probability model). We made two separate analyses using maximum likelihood—in the first, all branches of the phylogenetic tree were given an equal length (=1), while in the second, the length of branches was adjusted to reflect the estimated divergence dates between the species (Appendix A). We used primarily the dates from Kuhl et al. [29] to calibrate the deep divisions in the avian tree of life. However, that study was concentrated on higher-level relationships, and many species used in our study were not included. Therefore, many divergence dates for species that were missing from Kuhl et al. [29] were taken from TimeTree [41]. The estimated divergence dates from Kuhl et al. [29] are usually higher than in another recent comprehensive genetic analysis [28] or inferences based on the fossil record [42] but our tests indicate that such relatively small differences have a negligible effect on the obtained results.

## 3. Results

### 3.1. Development of the Vertebral Column in Podiceps cristatus

Ossification state 1. The osteogenesis of the vertebral column starts with the ossification of the bodies in five thoracic vertebrae (T3–T7) (Figure 5).

Ossification state 2. The ossification proceeds both anteriorly and posteriorly. In total, bodies of seven thoracic (T2–T8) and first four synsacral (S1–S4) vertebrae are ossified.

Ossification state 3. The ossification further spreads anteroposteriorly. The bodies of all thoracic (T1–T9), all synsacral, two posteriormost cervical vertebrae (C17–C18) and first caudal vertebrae are ossified (in total, 20 vertebrae). Additionally, the bodies of cervical vertebrae C2–C5 ossify; the ossification in the axis is very small (Figure 6 and Figure 7).

Ossification state 4. There are ossifications in the first seven cervical vertebrae (C1–C7) but, in the atlas, only the vertebral arches (unfused) are present. In the first four post-atlantal vertebrae (C2–C5), the vertebral arches begin to ossify. The bodies are present in the posterior cervical vertebrae (C14–C18), all thoracic and almost all synsacral vertebrae (in the studied specimen, the anterior caudals were not yet ossified, even though they were present in a specimen representing previous stage).

Ossification state 5. The bodies of all vertebrae from the first cervical to first caudal vertebrae are ossified, though the mid-cervicals are very poorly ossified. The vertebral arches are present in the first six cervical vertebrae (C1–C6), but the most posteriorly located are much more poorly developed.

Ossification state 6. The vertebral arches are present in all cervical vertebrae, though the more posterior ones are poorly ossified. Very small costal processes are visible in the vertebrae C3–C6. They are not fused to the body.

Ossification state 7. The ossification of the vertebral column extends further posteriorly. The vertebral arches are well developed, but still not fused to the body, in all cervical vertebrae (C1–C18). In contrast to the specimen representing previous stage, the costal processes were not ossified in the specimen from stage 7, demonstrating the presence of some variation.

Ossification state 8. The vertebral arches are fused in all cervicals and thoracics.

### 3.2. Development of the Vertebral Column in Podiceps grisegena

Ossification state 1. The bodies are well ossified in all cervical, thoracic, synsacral and most caudal vertebrae. The vertebral arches are well developed in the cervical and thoracic vertebrae. In the cervicals, the arches are already fused, but in more posteriorly located vertebrae, they are still separate, which indicates an anteroposterior order of the arch fusion (Figure 8).

Ossification state 2. The vertebral arches fuse in the remaining thoracic vertebrae.

### 3.3. Development of the Vertebral Column in Columba livia domestica

Hatchling. The bodies of the cervical vertebrae C3–C5 ossify from two lateral ossifications. There is only a single ossification in the axis. The ossifications in the axis and C3 are more pronounced than in the remaining vertebrae (Figure 9 and Figure 10).

1-day-old. The bodies are present in the cervical vertebrae C6–C10, but the more posteriorly located vertebrae are more poorly ossified (Figure 11).

2-day-old. There are ossifications in the bodies and vertebral arches in all cervical vertebrae. The degree of ossification decreases in more posteriorly located vertebrae. In C8–C12, the vertebral arches remain partly cartilaginous. The cervical vertebral arches remain separate in all vertebrae. In C3–C8 the caudal zygapophyses are more pronounced. In all cervical vertebrae except C12, the ossification centres for future costal processes appear ventrally to their corresponding vertebrae. The bodies are present in all thoracic vertebrae. The bodies of T3–T5 are slightly better ossified, which may indicate that the ossification starts there and proceeds bidirectionally. In T1, the vertebral arches are ossified. In T2–T6, the transverse processes begin to ossify. There are ossifications in the bodies of the synsacral vertebrae (S1–S10). The more posteriorly located ones are less well ossified (Figure 12).

4-day-old. In the cervical vertebrae, the vertebral arches begin to fuse, which indicates an anteroposterior order of arch fusion.

### 3.4. Ancestral State Reconstructions

Ancestral state reconstruction using the maximum parsimony suggests that the presence of two loci of the vertebral bodies ossification (one located in the cervical, the other in the thoracic vertebrae) is ancestral for most major avian clades—Neornithes, Palaeognathae, Neognathae and Neoaves (Figure 13). The thoracic locus seems to be lost in Psittacopasserae and possibly in Galloanserae (with either a few independent losses or a single loss and a few secondary re-evolutions). Almost the same result was also obtained in maximum likelihood analysis using equal branch lengths (=1) (Figure 14A, Appendix A). The analysis conducted on the temporally calibrated tree also gave similar results; however, it reconstructed the thoracic locus to be more likely ancestrally present in Galloanserae, Galliformes and Psittacopasserae, unlike the previous analyses (Figure 14B). The estimated probabilities for these clades are relatively low, so it is best to consider the results as ambiguous (Appendix A).

## 4. Discussion

The anteroposterior order of the ossification in the vertebral column has long been thought to be ancestral for amniotes [13,23]. Newer analyses confirm this assumption but indicate a somewhat more complex picture. Ancestrally, the vertebral bodies start ossifying from two loci in the column—a cervical one and a thoracic one (from which the ossification proceeds bidirectionally). The vertebral arches more closely conform to the anteroposterior sequence, with the ossification starting from one cervical locus in the majority of hitherto studied species [13]. The strictly anteroposterior order of ossification of the vertebral bodies and arches is present in hitherto studied crocodylians, the extant sister group of birds—two alligatorids, *Alligator mississippiensis* [43] and *Melanosuchus niger* [44]. Unfortunately, our knowledge of the skeletogenesis of the vertebral column in fossil birds is very incomplete. Many embryonic, perinatal and juvenile specimens are known in enantiornitheans (Enantiornithes) (e.g., [45,46,47,48]) and a few in ornithuromorphs [49,50]. However, enantiornitheans were a clade of superprecocial birds and they had a very well-ossified skeleton at the time of hatching. The skeleton presumably, like in other precocial birds (e.g., [16,18]), ossified rapidly, already during embryonic development. These specimens give us much information about the evolution of avian developmental sequences and, for example, the timing of neurocentral suture closure [45], but they represent relatively advanced developmental stages and are not informative regarding the order of ossification of the vertebral column. The same situation is also present with non-avian theropods. The embryos are known for several clades of maniraptorans closely related to birds, such as therizinosauroids [51], oviraptorosaurs (e.g., [15,52,53,54]) and troodontids [55]. However, the exact ossification sequence of the vertebral column cannot be deduced because the known specimens are not complete enough or represent near-hatching stages when the vertebrae (at least most of them) are already ossified. Additionally, embryos are very small and delicate fossils, and it is usually very difficult to determine whether, for example, the lack of some vertebrae represents their genuine absence or merely an incompleteness of the specimen. Therefore, the evolutionary history of the ossification patterns of the avian vertebral column must be reconstructed based on extant taxa alone (at least for now).

It was long thought that birds universally exhibit the typical anteroposterior pattern [16,18]. In particular, Starck [18] studied perinatal specimens of a wide range of birds and in none of them he found that ossification of posterior vertebrae preceded the ossification of more anteriorly located vertebrae. However, his study sample was rather small (at most five specimens per species, in most cases only one or two) and included perinates rather than embryos, so it left out the embryonic skeletal development. This was the probable reason for this misinterpretation. Further studies, using larger samples, indicated that the cervical vertebrae are not necessarily the first to ossify in birds (see ‘Literature Review’). The vertebral column in *Podiceps cristatus* was stated to ossify in a typical, anteroposterior sequence by Schinz and Zangerl [17], but their sample included only five specimens. Our data suggest that their conclusion about the development of the axial skeleton was probably a sampling artefact. The vertebral bodies ossify from two loci in *P. cristatus*—the cervical and thoracic. The same condition is also present in palaeognathous birds [30]. Although many parts of the phylogeny of Neoaves are still unresolved, the Podicipediformes seem to occupy a basal position within this clade (see review in Braun and Kimball [27]). It is thus possible that the ossification pattern observed in Palaeognathae and *P. cristatus* represents an ancestral state and the ‘typical’ anteroposterior order of ossification present in galloanserans [20,21] and many neoavian lineages [12,14,36] are convergent. The vertebral column in the pigeon ossifies with a pattern like the one observed in the grebe, with two ossification loci (cervical and thoracic). The cervical vertebrae, however, ossify before the thoracics, so it more closely conforms to the typical anteroposterior order. Our ancestral state reconstructions support the presence of two ossification loci in the vertebral column as an ancestral state for all major avian clades—Neornithes, Palaeognathae, Neognathae and Neoaves. They thus confirm the result of Verrière et al. [13], which was focused on amniotes in general.

Within extant birds, there is noticeable variation in both the number of ossification loci (one or two) and the direction of ossification in the vertebral column. However, there seems to be no clear correlation between these characteristics and eco-developmental strategies, precociality or altriciality. For example, in the precocial palaeognaths, the ossification of the vertebral column starts in the thoracic region, while the cervical locus appears later [30]. In two of the three members of the Anseriformes studied so far, which are also precocial, the ossification starts from a single cervical locus and proceeds posteriorly [20]. The same pattern of ossification as in anseriforms has also been observed in highly altricial parrots and passerines [12,14]. Therefore, it seems that the number of loci and direction of ossification does not have an adaptive significance and the observed variation is a result of the phylogeny.

Interestingly, in a previous article on the skeletal development of the pigeon [14], it was described that the vertebral column (specifically, the cervical vertebrae) starts ossifying already during embryonic development. In our sample, the hatchling had poorly ossified bodies only in the most anteriorly located cervicals. These differences between our sample and the specimens studied by Schinz and Zangerl [17] possibly suggest the variation in timing of ossification in the pigeon. This possibility cannot be excluded, as the pigeon is a highly variable species (e.g., [56]), and needs to be tested by studying the skeletal development in different breeds. The variation in trait growth, though not in the sequence of development, has recently been demonstrated for several chicken breeds [57].

The regionalisation of the vertebral column results primarily from the *Hox* gene expression [24,58,59]. These are regulatory genes that are expressed along the anteroposterior body axis. Cumulative effects of the expression of different *Hox* genes are the main determinants of the regionalisation of the vertebral column. For example, the transition between cervical and thoracic vertebrae, which is determined by the position of the pectoral girdle, depends on *Hoxb4* and *Hoxb9* which regulate the activity of *Tbx5*, the forelimb initiation gene [59]. However, the expression of homeotic genes may differ, depending on the activity of regulatory genes [60] or developmental origin of a given adult structure, e.g., from primaxial or abaxial mesoderm [3]. The ossification sequences are thus influenced by *Hox* gene expression. Interestingly, we observed that in *P. cristatus* the ossifications are not necessarily strictly correlated with the boundaries of vertebral column regions. The ossification of the bodies first developed in a locus located in anterior thoracic vertebrae and then spread posteriorly but also anteriorly into the cervical vertebrae. This suggests that the boundaries between different axial regions are not necessarily ‘obstacles’ for the spreading ossifications (irrespective of the direction of ossification, i.e., from anterior to posterior or from posterior to anterior). Further studies are needed to fully understand the relationships between homeotic genes and the direction and sequence of ossification in birds.

## 5. Conclusions

The ossification patterns of the vertebral column exhibit a relatively large variation in birds. In most currently studied species, the ossification of the vertebral bodies starts from two loci, one located in cervical vertebrae (from which it proceeds anteroposteriorly), the other in the thoracic vertebrae (from which it proceeds bidirectionally). The thoracic locus was probably lost at least two or three times, as it is absent in psittaciforms, passeriforms, anseriforms and at least one galliform species. The vertebral arches start ossifying from a single, cervical locus in the majority of hitherto studied species. The order of fusion between the arches was described in only a few species. In three of them (the galliform *Phasianus* sp. and the charadriiforms *Sterna hirundo* and *Chroicocephalus ridibundus*) the fusion appears to proceed bidirectionally from a single locus located in mid-cervical vertebrae. In *Podiceps grisegena* and *Columba livia* there is a single locus also located in the cervicals, but in the passeriform *Acrocephalus scirpaceus* there are three loci of fusion between the arches (located in anterior and posterior cervicals and mid-thoracics) which indicates a greater but very poorly explored variation within birds.

## Figures and Tables

**Figure 1 biology-11-00180-f001:**
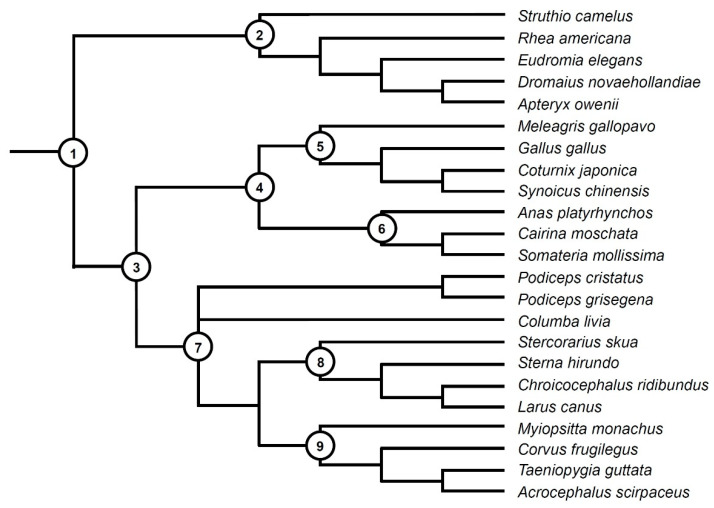
Phylogenetic relationships of the species discussed in the text. The topology of the dendrogram follows the ‘consensus phylogeny of birds’ from Braun and Kimball [27], while the interrelationships of the main clades follow primarily Prum et al. [28] and Kuhl et al. [29]. Codes: 1—Neornithes, 2—Palaeognathae, 3—Neognathae, 4—Galloanserae, 5—Galliformes, 6—Anseriformes, 7—Neoaves, 8—Charadriiformes, 9—Psittacopasserae.

**Figure 2 biology-11-00180-f002:**
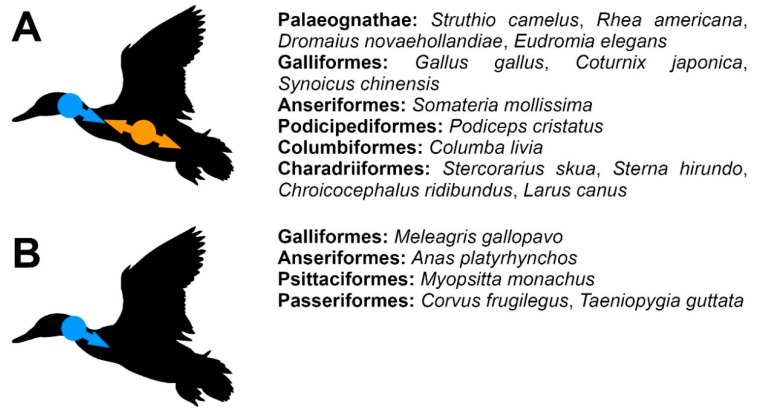
A schematic summary of the ossification patterns of the vertebral bodies in birds (**A**,**B**). Blue indicates cervical loci while orange indicates thoracic loci. The bird silhouette was taken from PhyloPic and is available in public domain.

**Figure 3 biology-11-00180-f003:**
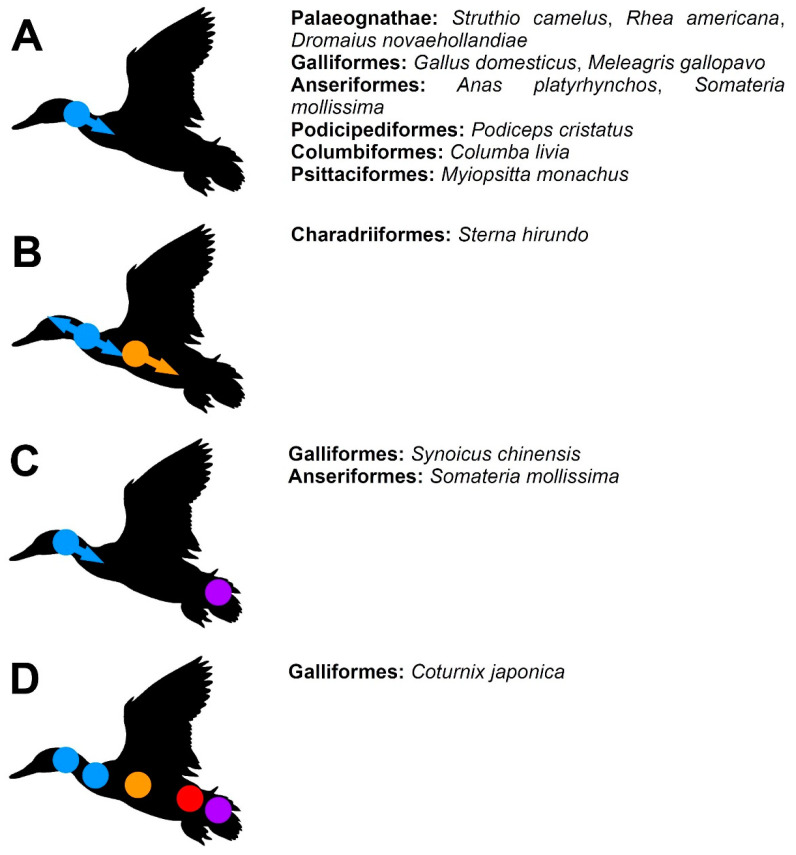
A schematic summary of the developmental patterns of the vertebral arches in birds (**A**–**D**). Ossification (appearance) of the vertebral arches. Blue indicates cervical loci, orange—thoracic loci, red—synsacral loci, purple—caudal loci.

**Figure 4 biology-11-00180-f004:**
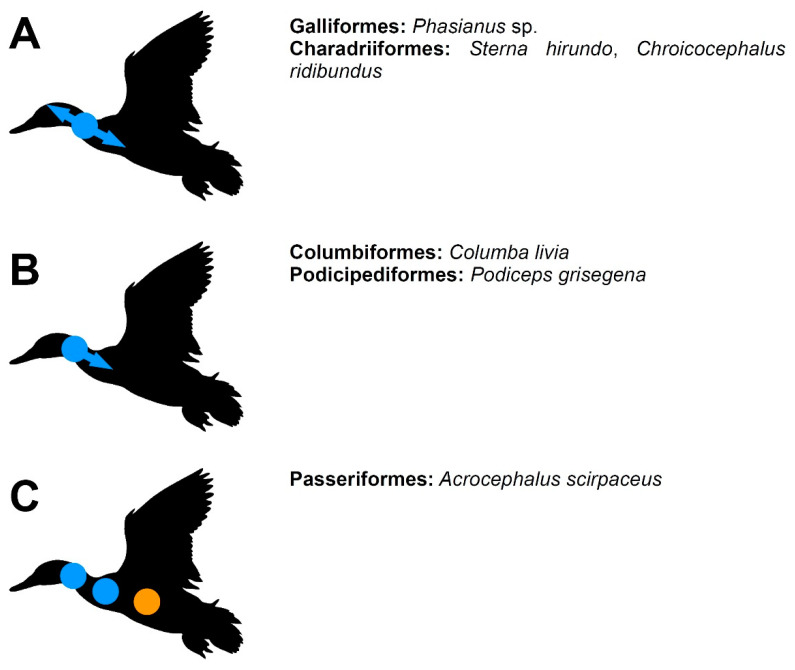
A schematic summary of developmental patterns in the vertebral arch fusion (**A**–**C**). Blue indicates cervical loci, orange—thoracic loci.

**Figure 5 biology-11-00180-f005:**
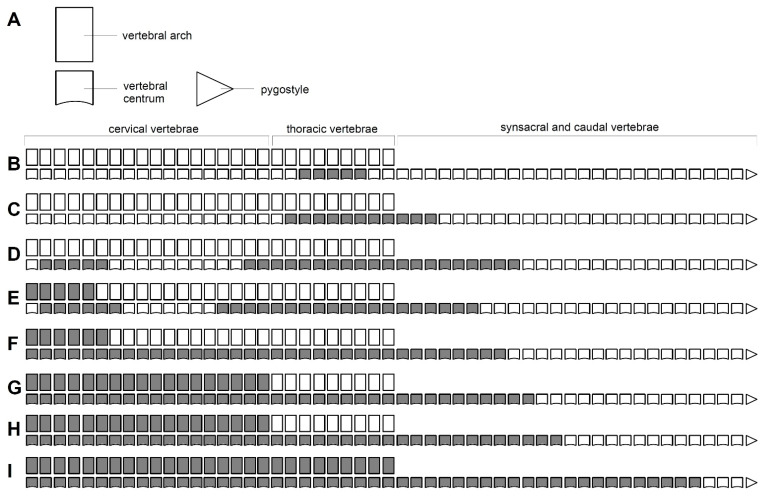
Ossification pattern of the vertebral column in *Podiceps cristatus*. (**A**) A schematic drawing of a single vertebra, showing the body and the vertebral arch. (**B**–**I**) Ossifications present in *P. cristatus* specimens, from state 1 (**A**) to state 8 (**I**). Grey colour indicates the presence of bone, while white indicates cartilage.

**Figure 6 biology-11-00180-f006:**
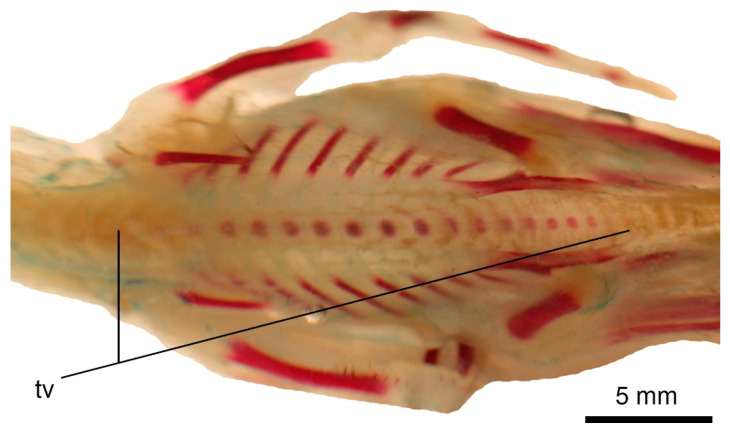
Ossification of the vertebral column in a state 3 specimen of *Podiceps cristatus*. The thoracic locus (with the ossification spreading into cervical and lumbosacral (synsacral) vertebrae) is seen on the photograph. Scale bar = 5 mm. tv—thoracic vertebrae.

**Figure 7 biology-11-00180-f007:**
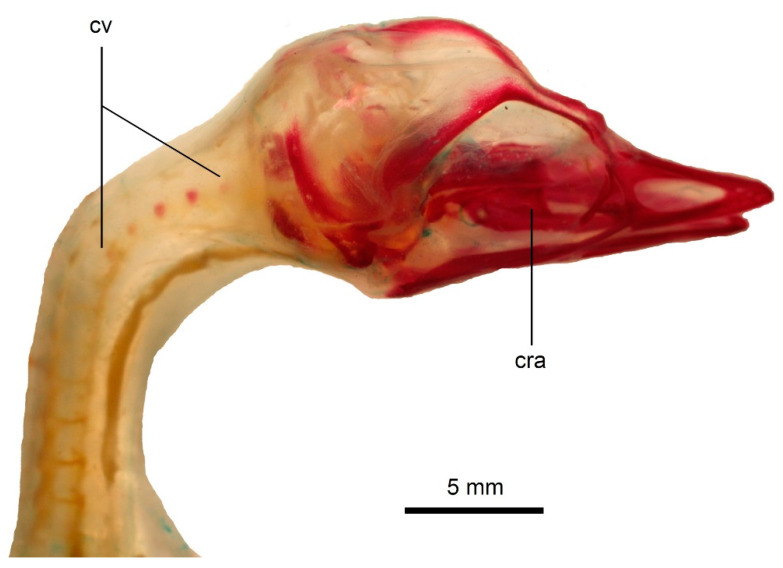
Ossification of the vertebral column in a state 3 specimen of *Podiceps cristatus*. The cervical locus is seen on the photograph. Scale bar = 5 mm. cra—cranium, cv—cervical vertebrae.

**Figure 8 biology-11-00180-f008:**
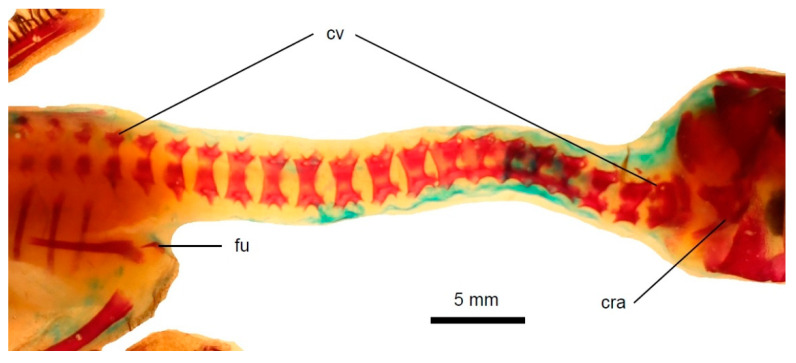
The neck and thorax of a neonatal *Podiceps grisegena*. The vertebral arches are fused in cervical vertebrae but still separate in the thoracic vertebrae. Scale bar = 5 mm. cra—cranium, cv—cervical vertebrae, fu—furcula.

**Figure 9 biology-11-00180-f009:**
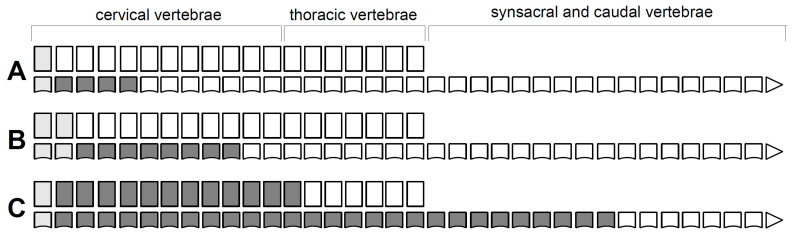
Ossification pattern of the vertebral column in *Columba livia*. (**A**–**C**) Ossifications present in studied specimens, from specimens at the time of hatching (**A**) to 3 days old (**C**). Dark grey indicates the presence of bone, while white indicates cartilage. The vertebrae that are missing in the specimen are highlighted by light grey. See Figure 5A for the legend.

**Figure 10 biology-11-00180-f010:**
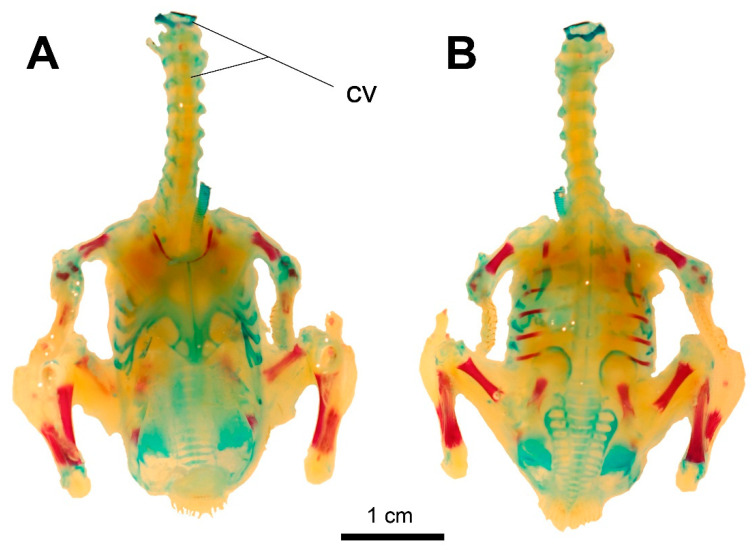
The state of ossification in a hatchling *Columba livia domestica*. (**A**) Ventral view. (**B**) Dorsal view. Scale bar = 1 cm. cv—cervical vertebrae.

**Figure 11 biology-11-00180-f011:**
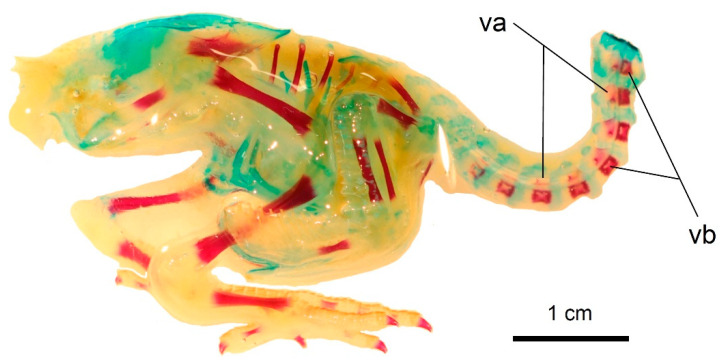
The state of ossification in a 1-day-old *Columba livia domestica* in lateral view. Scale bar = 1 cm. va—vertebral arches, vb—vertebral bodies.

**Figure 12 biology-11-00180-f012:**
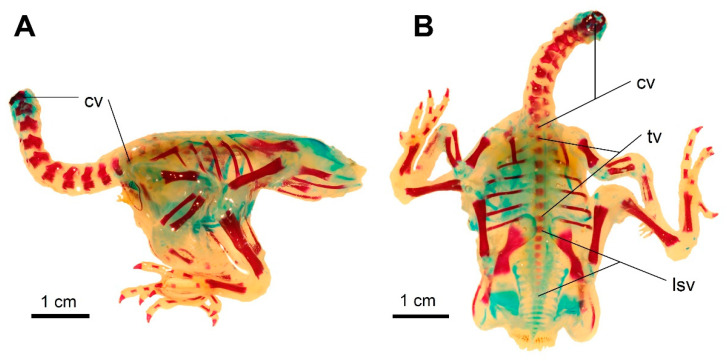
The state of ossification in a 2-day-old *Columba livia domestica*. (**A**) Lateral view. (**B**) Dorsal view. Scale bar = 1 cm. cv—cervical vertebrae, tv—thoracic vertebrae, lsv—lumbosacral (synsacral) vertebrae.

**Figure 13 biology-11-00180-f013:**
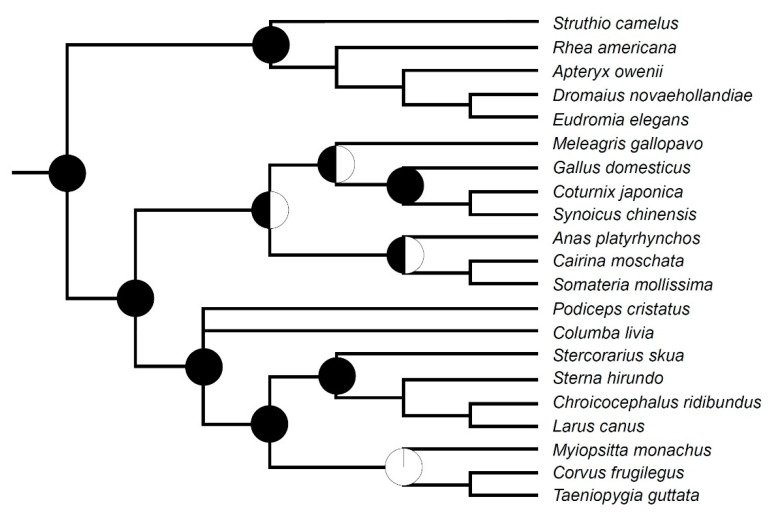
Parsimony-based ancestral state reconstructions of the ossification patterns of the vertebral body ossification. Black indicates the presence of a thoracic locus, while white indicates the lack thereof.

**Figure 14 biology-11-00180-f014:**
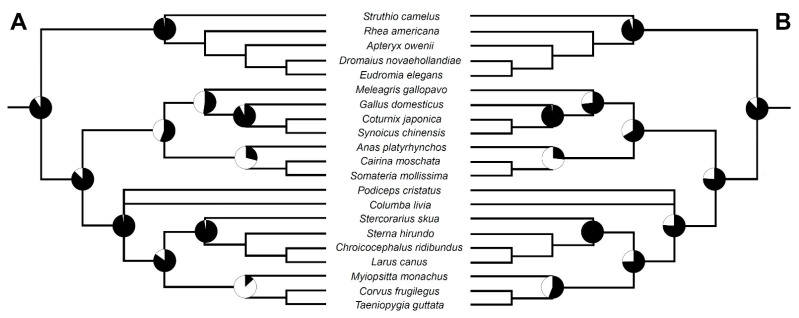
Maximum likelihood-based ancestral state reconstructions of the ossification patterns of the vertebral body ossification. Black indicates the presence of a thoracic locus, while white indicates the lack thereof. (**A**) Reconstructions based on the tree with equal branch length (=1). (**B**) Reconstructions based on the temporally calibrated tree. See Appendix A for the exact proportional likelihood values.

**Table 1 biology-11-00180-t001:** A summary of the ossification sequences of the vertebral bodies and arches in extant birds.

Species	Sequence	References
*Struthio camelus*	Vertebral bodies: thoracic, synsacral → cervical → caudal → pygostyle Vertebral arches: cervical → thoracic → synsacral	[30]
*Rhea americana*	Vertebral bodies: thoracic → cervical → synsacral Vertebral arches: cervical → thoracic → synsacral	[30]
*Dromaius novaehollandiae*	Vertebral bodies: cervical, thoracic → synsacral → caudal → pygostyle Vertebral arches: cervical → thoracic → synsacral	[30]
*Eudromia elegans*	Vertebral bodies: thoracic, synsacral → cervical Vertebral arches: cervical → thoracic → synsacral	[30]
*Meleagris gallopavo*	Vertebral bodies: cervical → thoracic → synsacral → caudal Vertebral arches: cervical → thoracic → synsacral	[21,31]
*Galllus domesticus*	Vertebral bodies: cervical → thoracic → synsacral → caudal Vertebral arches: cervical → thoracic → synsacral	[17,21]
*Coturnix japonica*	Vertebral bodies: cervical → thoracic → synsacral → caudal Vertebral arches: cervical → thoracic → synsacral	[21,32,33]
*Synoicus chinensis*	Vertebral bodies: anterior cervical, thoracic → mid- and posterior cervical, synsacral → caudal Vertebral arches: cervical → thoracic → synsacral	[34]
*Anas platyrhynchos*	Vertebral bodies: cervical → thoracic, synsacral → caudal → pygostyle Vertebral arches: cervical → thoracic → synsacral	[12,20]
*Cairina moschata*	Vertebral bodies: cervical, thoracic, synsacral → caudal → pygostyle Vertebral arches: cervical → thoracic	[20]
*Somateria mollissima*	Vertebral bodies: cervical → thoracic, synsacral → caudal → pygostyle Vertebral arches: cervical → thoracic → synsacral	[20]
*Podiceps cristatus*	Vertebral bodies: thoracic → synsacral → cervical → caudal → pygostyle Vertebral arches: cervical → thoracic → synsacral	this work
*Columba livia domestica*	Vertebral bodies: cervical → thoracic, synsacral → caudal Vertebral arches: cervical → thoracic	this work
*Sterna hirundo*	Vertebral bodies: cervical, thoracic, synsacral → caudal Vertebral arches: cervical → thoracic or thoracic → cervical	[22]
*Stercorarius skua*	Vertebral bodies: cervical, thoracic → synsacral → caudal Vertebral arches: cervical → thoracic	[22,35]
*Chroicocephalus ridibundus*	Vertebral bodies: cervical → thoracic → synsacral → caudal Vertebral arches: cervical → thoracic	[22,36,37]
*Larus canus*	Vertebral bodies: cervical, thoracic, synsacral → caudal Vertebral arches: thoracic → cervical	[22,37]
*Myiopsitta monachus*	Vertebral bodies: cervical → thoracic → synsacral, caudal	[14]
*Taeniopygia guttata*	Vertebral bodies: cervical → thoracic → synsacral, caudal	[12]
*Corvus frugilegus*	Vertebral bodies: cervical → thoracic → synsacral → caudal	[36]

## Data Availability

All data generated by this study are available in this manuscript and the accompanying Appendix A.

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
