# Peer review of "Phylogenetic Diversity of Ossification Patterns in the Avian Vertebral Column: A Review and New Data from the Domestic Pigeon and Two Species of Grebes"

_biology, 2022, doi:10.3390/biology11020180_

Round 1
Reviewer 1 Report
In this manuscript, the authors review the available information about the development of the vertebral column in birds, describe the ossification pattern of the vertebral column in three Neoavian species (the pigeon -Columba livia domestica- and two grebes -Podiceps cristatus and Podiceps grisegena-), and reconstruct the evolutionary history of the ossification patterns of the vertebral centra.
Although the scientific question is original and well defined, I think there are some mayor issues that must be addressed:
- I believe that 12 specimens of each Podiceps and 5 specimens of Columba is a very small sample for a developmental study. I know it is difficult to get specimens, but in this case this are species categorized as least concerned, and even more, the domestic pigeon is considered as a worldwide pest. Furthermore, in the discussion section, the authors argue that this type of small sample could lead other authors conducting similar researches to misinterpretations or erroneous conclusions (lines 411-422).
- Why did the authors choose the species they describe? It seems that those species were the ones they had available. Also, why do the authors analyze two species of grebes? The species´ choice must be based on a research question or on an objective. Moreover, the pigeon is a highly variable species that needs to be tested by studying different breeds (as the authors state in lines 452-454). And given the small sample of species (21) analyzed within the phylogeny, the results from the ancestral state reconstruction could be biased by the use of this variable species.
- Why do the authors include the species Podiceps grisegena and Acrocephalus scirpaceus in the phylogeny but do not include them in the ancestral state reconstruction? (line 291).
- The authors use a combined phylogeny by taking data from three different phylogenies and use estimated divergence dates from three authors. This is a usual procedure, but the authors must detail and justify the reasons for this decision, especially because they analyzed a small sample of species (21).
Minor issues:
- Simple Summary: Neoaves includes all living birds except the Palaeognathae and the Galloanserae (Galliformes + Anseriformes) (not only the fowl).
- Simple Summary: the authors state “In the pigeon, the neck vertebrae were the first to ossify but in the grebe, the thoracic vertebrae ossified earlier”. Did you mean that in the grebe the first vertebrae to ossify are the thoracic ones? Because in a comparative developmental context “earlier” denotes changes in time, but not changes in the ossification sequence. I mean, thoracic vertebrae could ossify earlier in the grebe with respect of the pigeon, but ossify second in order in the grebe (with the neck vertebrae ossifying even earlier).
- Introduction: the quote of the phrase “They also help us interpret fossil embryos…” (line 60) is from Balanoff & Rowe 2007 (not Mitgutsch et al. 2011).
- Introduction: I suggest changing the definition of precocial bird from “they hatch at a relatively advanced developmental stage” (line 65) to “independent hatchlings”, because it can be confused with the H&H developmental stages.
- Introduction: pigeons (Columbidae) are altricial type 1 according to Starck (1993) (lines 86-87).
- Literature Review: this section is unnecessarily long. It could be summarized in a table and it should be included in the M&M section (or even as supplementary information).
- Literature Review: being a work of ossification patterns in a comparative context, I recommend for descriptions the use of the H&H developmental stages (if not possible, the ossification sequence) and not days since birds differ markedly in incubation time and could be quite confusing.
- M&M: a staging table for the pigeon is available: Olea & Sandoval (2012). Embryonic development of Columba livia (Aves: Columbiformes) from an altricial-precocial perspective. Revista Colombiana de Ciencias Pecuarias 25:3-13).
- M&M: the authors state “The identification and nomenclature of anatomical structures follow primarily Baumel and Witmer” (lines 286-287). Thus, they should use the Latin names and avoid terms such as “spine” or “neck vertebrae”.
- Results: I suggest showing the results of the ancestral state reconstruction in a cladogram, not in a table.
- Discussion: the authors state “The vertebral column in the pigeon ossifies much more rapidly but the pattern of ossification is similar to the one observed in the grebe”. This is an interesting pattern and the authors should discuss it more, since pigeons are altricial and grebes are precocial and a delayed pattern of ossification in the pigeon would be expected.
- Discussion: do the authors mean “They thus confirm the result of Verrière et al. [24] with a taxonomically SMALLER sample of birds”? (lines 434-435).
Author Response
Dear Editors and Referees,
first and foremost, please accept our 'thank you' for your time and effort in reviewing our article. We greatly appreciate your input. We did our best to improve the article based on your suggestions. In a few cases in which we do not agree with your opinions, we did our best to provide a convincing explanation. Below is our point-by-point response to the comments made by the Referees (our responses are in bold).
Sincerely,
Tomasz Skawiński
REVIEWER 1
Comments and Suggestions for Authors
In this manuscript, the authors review the available information about the development of the vertebral column in birds, describe the ossification pattern of the vertebral column in three Neoavian species (the pigeon -Columba livia domestica- and two grebes -Podiceps cristatus and Podiceps grisegena-), and reconstruct the evolutionary history of the ossification patterns of the vertebral centra.
Although the scientific question is original and well defined, I think there are some mayor issues that must be addressed:
- I believe that 12 specimens of each Podiceps and 5 specimens of Columba is a very small sample for a developmental study. I know it is difficult to get specimens, but in this case this are species categorized as least concerned, and even more, the domestic pigeon is considered as a worldwide pest. Furthermore, in the discussion section, the authors argue that this type of small sample could lead other authors conducting similar researches to misinterpretations or erroneous conclusions (lines 411-422).
We agree that our sample size would be small for a 'typical' study in which the development of the skeleton is described. First, we were able to increase the sample size for P. grisegena to 18 specimens. Second, please note that we were interested only in a relatively short period of the skeletogenesis during which the vertebral column ossifies. This includes the final stages (39+ in the Hamburger and Hamilton staging table) of the embryonic development in Podiceps cristatus and the first few days of postnatal development in Columba livia. These samples were sufficient to allow us to observe the spreading of ossifications in the vertebral column and show a coherent and consistent developmental pattern (we were critical of studies in which the conclusions about the direction of ossification were based on a sample of one or two specimens, which are of course based on at most two developmental stages, so much lower than in our article).
Both species of grebes are indeed listed as 'Least Concern' by the IUCN but they are quite rare and both are strictly protected in Poland and getting permissions to collect wild individuals (including dead ones) is difficult. Therefore, we decided to utilise the specimens that were collected previously and used also in other studies.
- Why did the authors choose the species they describe? It seems that those species were the ones they had available. Also, why do the authors analyze two species of grebes? The species´ choice must be based on a research question or on an objective.
Our choice of the species was indeed partly based on the availability of the material. However, both the grebes and the pigeon are – according to recent comprehensive phylogenetic analyses (reviewed by Braun & Kimball 2021) – members of the early-diverging lineages of the Neoaves. Because the ossification patterns of the vertebral column have so far been described only for deeply-nested lineages, the use of more distantly related species is crucial for more meaningful reconstructions of the evolutionary history. We made a more detailed justification in the 'Introduction'.
Moreover, the pigeon is a highly variable species that needs to be tested by studying different breeds (as the authors state in lines 452-454). And given the small sample of species (21) analyzed within the phylogeny, the results from the ancestral state reconstruction could be biased by the use of this variable species.
Our study suggests that the pigeon exhibits the ossification pattern of the vertebral column which is quite widespread in birds and probably ancestral for the clade (this result was obtained even in the analyses which did not include this species; e.g. Verriere et al. 2021). Unfortunately, the intraspecific variation in development is very poorly studied in general (including the species in which the skeletal development is much better known such as the quail).
- Why do the authors include the species Podiceps grisegena and Acrocephalus scirpaceus in the phylogeny but do not include them in the ancestral state reconstruction? (line 291).
The figure is supposed to show the relationships between all species discussed in the text but in P. grisegena and A. scirpaceus the ossification patterns of the vertebral centra and arches were not known so they were not included in the ancestral state reconstructions.
- The authors use a combined phylogeny by taking data from three different phylogenies and use estimated divergence dates from three authors. This is a usual procedure, but the authors must detail and justify the reasons for this decision, especially because they analyzed a small sample of species (21).
We made a more detailed justification in the section 2.3., 'Ancestral State Reconstruction'. Hopefully, it is sufficient. In brief, we had to combine the data because many of the species which were studied in terms of development, were not included in phylogenetic analyses, so the information about their phylogenetic position or divergence dates had to be taken from different sources.
Minor issues:
- Simple Summary: Neoaves includes all living birds except the Palaeognathae and the Galloanserae (Galliformes + Anseriformes) (not only the fowl).
'Fowl' is a term that refers to galloanserans in general and includes both landfowl (galliforms) and waterfowl (anseriforms) but we decided to rephrase the sentence to remove any possible ambiguities.
- Simple Summary: the authors state “In the pigeon, the neck vertebrae were the first to ossify but in the grebe, the thoracic vertebrae ossified earlier”. Did you mean that in the grebe the first vertebrae to ossify are the thoracic ones? Because in a comparative developmental context “earlier” denotes changes in time, but not changes in the ossification sequence. I mean, thoracic vertebrae could ossify earlier in the grebe with respect of the pigeon, but ossify second in order in the grebe (with the neck vertebrae ossifying even earlier).
We slightly rephrased the sentence. Hopefully, it is clear now.
- Introduction: the quote of the phrase “They also help us interpret fossil embryos…” (line 60) is from Balanoff & Rowe 2007 (not Mitgutsch et al. 2011).
We believe that citing Mitgutsch et al. is justified to support this sentence but we added some citations referring specifically to studying fossil embryos.
- Introduction: I suggest changing the definition of precocial bird from “they hatch at a relatively advanced developmental stage” (line 65) to “independent hatchlings”, because it can be confused with the H&H developmental stages.
Done.
- Introduction: pigeons (Columbidae) are altricial type 1 according to Starck (1993) (lines 86-87).
We added this information to the article.
- Literature Review: this section is unnecessarily long. It could be summarized in a table and it should be included in the M&M section (or even as supplementary information).
We added two figures and a table in an attempt to summarise this review. However, the ossification patterns of the vertebral column are very diverse in birds and need thorough description and we failed to make the text shorter without losing information (which, in our opinion, is valuable). We followed the suggestion and moved the section to 'Materials and Methods'.
- Literature Review: being a work of ossification patterns in a comparative context, I recommend for descriptions the use of the H&H developmental stages (if not possible, the ossification sequence) and not days since birds differ markedly in incubation time and could be quite confusing.
We used the H&H stages wherever possible. However, some previous authors used different proxies for developmental stage (usually, the absolute age of the specimen). Also, the advanced developmental stages in the H&H table (40+) span several days in many species, so it is useful to indicate differences between, for example early stage 40+ and late stage 40+.
- M&M: a staging table for the pigeon is available: Olea & Sandoval (2012). Embryonic development of Columba livia (Aves: Columbiformes) from an altricial-precocial perspective. Revista Colombiana de Ciencias Pecuarias 25:3-13).
We are aware of this publication. However, our study was based on hatchlings so the embryonic staging table is of little utility. Therefore, we used the absolute age of the hatchlings as a proxy for their developmental stage.
- M&M: the authors state “The identification and nomenclature of anatomical structures follow primarily Baumel and Witmer” (lines 286-287). Thus, they should use the Latin names and avoid terms such as “spine” or “neck vertebrae”.
We believe that using English terms (for example, 'vertebral column' rather than 'columna vertebralis') in an article that is written in English makes the article easier to read so we decided to retain them (we added a disclaimer in the 'Material and methods'). The vernacular terms 'spine' and 'neck vertebrae' are currently used only in the Simple Summary which is aimed at wider audience and should use as little jargon as possible.
- Results: I suggest showing the results of the ancestral state reconstruction in a cladogram, not in a table.
Done. We added the requested figures and moved the table to the supplemental material.
- Discussion: the authors state “The vertebral column in the pigeon ossifies much more rapidly but the pattern of ossification is similar to the one observed in the grebe”. This is an interesting pattern and the authors should discuss it more, since pigeons are altricial and grebes are precocial and a delayed pattern of ossification in the pigeon would be expected.
Unfortunately, the cited sentence was poorly phrased and misleading – we did not mean to indicate that the vertebral column in the pigeon starts to ossify relatively earlier than in the grebe (which is not true as it ossifies near the time of hatching rather than during embryonic development). We rephrased the sentence in question.
- Discussion: do the authors mean “They thus confirm the result of Verrière et al. [24] with a taxonomically SMALLER sample of birds”? (lines 434-435).
We slightly rephrased the sentence. Hopefully, it is clear now.
Reviewer 2 Report
This study by Skawinski et al. review ossification patterns in the avian spinal column, identify previously underdetermined patterns in three members of Neoaves, and perform phylogenetic and evolutionary analyses to reconstruct the ancestral avian state for spinal ossification. Whilst I am by no means an expert in avian ossification patterns, I found this study to well designed, extensively researched and well written, including a literature review to accompany their own data. Though overall their data reaffirmed a similar ancestral state to that previously described, the addition of new taxonomic sampling strengthens previous findings, and places them into a broader evolutionary context.
My only critique for this manuscript is that the volume of information surmised in the Section 2 literature review was extremely dense and descriptive, which could be, at times, difficult to follow. As this section largely constitutes a review of the various ossification patterns observed throughout different species and clades of avians, I suggest that this section be accompanied by an additional summarized diagram that provides an illustrative overview of the observed ossification patterns in a phylogenetic context. This will greatly enhance the readability and significance of this extensive review, diagrammatically support the supplied information, and point out key areas where knowledge is lacking. A similar suggestion can be made for the results or discussion section of the manuscript. An additional figure could help to integrate the ossification patterns obtained in this study into the greater avian phylogeny. Before publication, I suggest that the authors include one of these figure variations to place their data into a broader context.
Similarly, the authors may determine a way to include data for each of the three species examined in figures 2-7, as there is only figures to support data for Podiceps cristatus and Columba livia domestica, but not Podiceps grisegena, despite with reduced sampling. However, while this might enhance the presentation of the data generated, this is not necessary and more of a suggestion.
Overall, this was a well written manuscript and adds new evolutionary insights into avian ossification patterns. With the addition of an additional summative figure, I support this manuscript for publication.
Minor comments:
Lines 456-472 should additionally include discussion from (Moreau et al., 2019), which examined vertebral boundaries and HOX genes between different avian species
Moreau, C., Caldarelli, P., Rocancourt, D., Roussel, J., Denans, N., Pourquie, O., & Gros, J. (2019). Timed Collinear Activation of Hox Genes during Gastrulation Controls the Avian Forelimb Position. Current Biology, 29(1), 35-50.e4. https://doi.org/10.1016/j.cub.2018.11.009
Author Response
Dear Editors and Referees,
first and foremost, please accept our 'thank you' for your time and effort in reviewing our article. We greatly appreciate your input. We did our best to improve the article based on your suggestions. In a few cases in which we do not agree with your opinions, we did our best to provide a convincing explanation. Below is our point-by-point response to the comments made by the Referees (our responses are in bold).
Sincerely,
Tomasz Skawiński
REVIEWER 2
Comments and Suggestions for Authors
This study by Skawinski et al. review ossification patterns in the avian spinal column, identify previously underdetermined patterns in three members of Neoaves, and perform phylogenetic and evolutionary analyses to reconstruct the ancestral avian state for spinal ossification. Whilst I am by no means an expert in avian ossification patterns, I found this study to well designed, extensively researched and well written, including a literature review to accompany their own data. Though overall their data reaffirmed a similar ancestral state to that previously described, the addition of new taxonomic sampling strengthens previous findings, and places them into a broader evolutionary context.
My only critique for this manuscript is that the volume of information surmised in the Section 2 literature review was extremely dense and descriptive, which could be, at times, difficult to follow. As this section largely constitutes a review of the various ossification patterns observed throughout different species and clades of avians, I suggest that this section be accompanied by an additional summarized diagram that provides an illustrative overview of the observed ossification patterns in a phylogenetic context. This will greatly enhance the readability and significance of this extensive review, diagrammatically support the supplied information, and point out key areas where knowledge is lacking. A similar suggestion can be made for the results or discussion section of the manuscript. An additional figure could help to integrate the ossification patterns obtained in this study into the greater avian phylogeny. Before publication, I suggest that the authors include one of these figure variations to place their data into a broader context.
We are aware of the fact that the literature review is very descriptive and quite 'dry'. However, these data are very difficult to present in a more accessible form. As suggested, we made a few figures and a table which attempt to summarise this section.
Similarly, the authors may determine a way to include data for each of the three species examined in figures 2-7, as there is only figures to support data for Podiceps cristatus and Columba livia domestica, but not Podiceps grisegena, despite with reduced sampling. However, while this might enhance the presentation of the data generated, this is not necessary and more of a suggestion.
We have added the requested figure.
Overall, this was a well written manuscript and adds new evolutionary insights into avian ossification patterns. With the addition of an additional summative figure, I support this manuscript for publication.
Minor comments:
Lines 456-472 should additionally include discussion from (Moreau et al., 2019), which examined vertebral boundaries and HOX genes between different avian species
Moreau, C., Caldarelli, P., Rocancourt, D., Roussel, J., Denans, N., Pourquie, O., & Gros, J. (2019). Timed Collinear Activation of Hox Genes during Gastrulation Controls the Avian Forelimb Position. Current Biology, 29(1), 35-50.e4. https://doi.org/10.1016/j.cub.2018.11.009
We have added a brief discussion about this publication.
Reviewer 3 Report
The authors review the diversity of ossification patterns in the avian vertebral column and supplement the paper with new data about three species of birds. Current literature discusses the existence of two sites from which the ossification of the spine starts in birds. It is the cervical and thoracic position. The authors of this paper reviewed the literature thoroughly under Introduction. The analysis of ancestral state reconstruction revealed that there were ancestrally two sites of ossifications. The thoracic site was secondarily lost in several groups of Neognathae.
I consider this manuscript interesting. Nevertheless, I have several suggestions on how to improve the paper. From my point of view, the most important is the presentation of the results. I think that it will be better to show the ancestral state reconstruction on the tree rather than in the table. Another thing is that the proper age of the samples is not known. This makes it impossible to compare the results between studied samples and species, but it is also not comparable for the following meta-analysis. The authors should at least explain why it does not matter. Please clarify also the choice of studied species. Moreover, I will greatly appreciate the more wide phylogenetic approach. The authors should at least mention, what is the pattern of ossification in related groups, e.g., some fossil avian species and other groups from Archelosauria clade).
Here are other rather minor suggestions:
line 62: Birds are not one of the most species-rich lineage of tetrapods but are the most species-rich lineage of tetrapods.
line 241: double and at the end of the line, something is missing…
line 280: The procedure was according to Dingerkus and Uhler with slight modifications. You should describe how.
line 311: the name of the bird should be in italics
line 314: In the description of Figure 2 A) I miss the pygostyle
line 316 and lower: How did you classify ossification states? Is it your classification? Alternatively, is it commonly used in developmental studies? Are those ossification states comparable between studied species? Please clarify.
line 327: remove redundant bracket
line 350: the name of the bird should be in italics
line 351: Do you think it is necessary to show those data? I think that the sampled specimens were too old, and it does not bring new information about the ossification.
line 354: should be: which indicates an anteroposterior order?
line 357: the name of the bird should be in italics
Section 4.3: I will greatly appreciate the schema of ossification of Columba livia in the same form as for Podiceps cristatus (Figure 2). Is it possible?
Figure 6: I miss the proper description of the figure. I guess va are vertebral arches? vc are cervical vertebrae? – should be cv
Figures 6 and 7: Why are the claws of the pigeon red? You mentioned under Material and methods that red coloured parts of the body are bones after ossification
Line 406 and 407: I miss the references for the first and second sentences.
Author Response
Dear Editors and Referees,
first and foremost, please accept our 'thank you' for your time and effort in reviewing our article. We greatly appreciate your input. We did our best to improve the article based on your suggestions. In a few cases in which we do not agree with your opinions, we did our best to provide a convincing explanation. Below is our point-by-point response to the comments made by the Referees (our responses are in bold).
Sincerely,
Tomasz Skawiński
REVIEWER 3
Comments and Suggestions for Authors
The authors review the diversity of ossification patterns in the avian vertebral column and supplement the paper with new data about three species of birds. Current literature discusses the existence of two sites from which the ossification of the spine starts in birds. It is the cervical and thoracic position. The authors of this paper reviewed the literature thoroughly under Introduction. The analysis of ancestral state reconstruction revealed that there were ancestrally two sites of ossifications. The thoracic site was secondarily lost in several groups of Neognathae.
I consider this manuscript interesting. Nevertheless, I have several suggestions on how to improve the paper. From my point of view, the most important is the presentation of the results. I think that it will be better to show the ancestral state reconstruction on the tree rather than in the table.
Done. We added the requested figures and moved the table to the supplemental material.
Another thing is that the proper age of the samples is not known. This makes it impossible to compare the results between studied samples and species, but it is also not comparable for the following meta-analysis. The authors should at least explain why it does not matter.
We disagree that the fact that the exact age of the specimens is usually not known makes it impossible to compare the data between species. We were interested in analysing the developmental sequences which can be observed even without any information about the exact ages of the studied specimens. Also, the age is unknown in the great majority of studies in which the specimens are collected from the wild.
Please clarify also the choice of studied species.
We have added a short justification in the 'Introduction'.
Moreover, I will greatly appreciate the more wide phylogenetic approach. The authors should at least mention, what is the pattern of ossification in related groups, e.g., some fossil avian species and other groups from Archelosauria clade).
Unfortunately, the data on the ossification patterns of the vertebral column (the aspects that we were attempting to explore in this article) are basically unknown in dinosaurs and fossil birds; that is why we initially did not discuss them in the text. However, we have now added a short discussion on the fossil record and patterns seen in other archosaurs.
Here are other rather minor suggestions:
line 62: Birds are not one of the most species-rich lineage of tetrapods but are the most species-rich lineage of tetrapods.
According to the IOC World Bird List (v. 11.2) there are 10,912 extant species of birds while there are 11,302 species of squamates (The Reptile Database). Given that the number of species varies as the new species are being described and some names are synonymised with others, we believe that it is safer to state that 'birds are one of the most species-rich lineage of tetrapods'.
line 241: double and at the end of the line, something is missing…
One 'and' removed.
line 280: The procedure was according to Dingerkus and Uhler with slight modifications. You should describe how.
The main deviation from the original procedure – the reduced amount of glacial acetic acid in the alcian blue staining solution – was already described in the text. Other modifications were minimal and concerned, for example, some bigger specimens needed more time in the staining or digesting solutions than smaller specimens. Such minor modifications are usually not described in other articles that use this method and we also decided to omit them.
line 311: the name of the bird should be in italics
line 350: the name of the bird should be in italics
line 357: the name of the bird should be in italics
The species names should be written in a different font than the rest of the text. The heading is written in italics so so we have decided to write the species name in non-italics in this context.
line 314: In the description of Figure 2 A) I miss the pygostyle
We have now corrected the description.
line 316 and lower: How did you classify ossification states? Is it your classification? Alternatively, is it commonly used in developmental studies? Are those ossification states comparable between studied species? Please clarify.
We tried to clarify that in the 'Material and methods' section. Hopefully, it is more clear now.
line 327: remove redundant bracket
Fixed.
line 351: Do you think it is necessary to show those data? I think that the sampled specimens were too old, and it does not bring new information about the ossification.
We do not have a strong opinion in this case but we decided to tentatively retain these data because – although not a direct focus of this article – they ma be useful to some other authors studying skeletal development.
line 354: should be: which indicates an anteroposterior order?
Yes, thanks for pointing this. It is now corrected.
Section 4.3: I will greatly appreciate the schema of ossification of Columba livia in the same form as for Podiceps cristatus (Figure 2). Is it possible?
Done.
Figure 6: I miss the proper description of the figure. I guess va are vertebral arches? vc are cervical vertebrae? – should be cv
We have now corrected the description.
Figures 6 and 7: Why are the claws of the pigeon red? You mentioned under Material and methods that red coloured parts of the body are bones after ossification
These are the ungual phalanges which are covered by horny sheath to form the claw.
Line 406 and 407: I miss the references for the first and second sentences.
The references have been added.
Round 2
Reviewer 1 Report
The authors have addressed all my suggestions and properly argued their responses. I only have one suggestion regarding the small sampling. Although it is always a challenge to find a balance between the number of specimens that are feasible to obtain and the number of specimens that are necessary to test a hypothesis, I still believe that 5 specimens of the dove is scarce. Here the authors state that the samples they used were sufficient to allow them to observe a coherent and consistent developmental ossification pattern in the vertebral column. But, they also argue in the discussion section that the conclusions reached by other authors (i.e., Starck 1996, Schinz and Zangerl 1937) about the development of the axial skeleton could be biased by the small sampling they used. Perhaps it would be a good idea to first explain in the M&M section that the sample that was used (although small) was sufficient to obtain the expected results and to lay the groundwork for future studies. And second, to rephrase the discussion section (lines 486-496) so it doesn't argue against the also small sample used in this paper. Beyond this, I believe this is an interesting article with an original scientific question.
Author Response
Dear Editors and Referee,
thank you very much for your further comments. Below we respond (in bold) to the comments.
Sincerely,
Tomasz Skawiński
REVIEWER 1
The authors have addressed all my suggestions and properly argued their responses. I only have one suggestion regarding the small sampling. Although it is always a challenge to find a balance between the number of specimens that are feasible to obtain and the number of specimens that are necessary to test a hypothesis, I still believe that 5 specimens of the dove is scarce. Here the authors state that the samples they used were sufficient to allow them to observe a coherent and consistent developmental ossification pattern in the vertebral column. But, they also argue in the discussion section that the conclusions reached by other authors (i.e., Starck 1996, Schinz and Zangerl 1937) about the development of the axial skeleton could be biased by the small sampling they used. Perhaps it would be a good idea to first explain in the M&M section that the sample that was used (although small) was sufficient to obtain the expected results and to lay the groundwork for future studies. And second, to rephrase the discussion section (lines 486-496) so it doesn't argue against the also small sample used in this paper. Beyond this, I believe this is an interesting article with an original scientific question.
Thank you for this suggestion. We agree with it and have added a note to the 'Materials and Methods' section and have slightly modified the mentioned part of the 'Discussion'.